# GamaComet: A Deep Learning-Based Tool for the Detection and Classification of DNA Damage from Buccal Mucosa Comet Assay Images

**DOI:** 10.3390/diagnostics12082002

**Published:** 2022-08-18

**Authors:** Edgar Anarossi, Ryna Dwi Yanuaryska, Sri Mulyana

**Affiliations:** 1Department of Computer Science and Electronics, Faculty of Mathematics and Natural Sciences, Universitas Gadjah Mada, Yogyakarta 55281, Indonesia; 2Division of Information Science, Institute for Research Initiatives, Nara Institute of Science and Technology, Ikoma 630-0192, Japan; 3Department of Radiology Dentomaxillofacial, Faculty of Dentistry, Universitas Gadjah Mada, Yogyakarta 55281, Indonesia

**Keywords:** comet assay image, buccal mucosa, Faster R-CNN, detection and classification, DNA damage

## Abstract

Comet assay is a simple and precise method to analyze DNA damage. Nowadays, many research studies have demonstrated the effectiveness of buccal mucosa cells usage in comet assays. However, several software tools do not perform well for detecting and classifying comets from a comet assay image of buccal mucosa cells because the cell has a lot more noise. Therefore, a specific software tool is required for fully automated comet detection and classification from buccal mucosa cell swabs. This research proposes a deep learning-based fully automated framework using Faster R-CNN to detect and classify comets in a comet assay image taken from buccal mucosa swab. To train the Faster R-CNN model, buccal mucosa samples were collected from 24 patients in Indonesia. We acquired 275 comet assay images containing 519 comets. Furthermore, two strategies were used to overcome the lack of dataset problems during the model training, namely transfer learning and data augmentation. We implemented the proposed Faster R-CNN model as a web-based tool, GamaComet, that can be accessed freely for academic purposes. To test the GamaComet, buccal mucosa samples were collected from seven patients in Indonesia. We acquired 43 comet assay images containing 73 comets. GamaComet can give an accuracy of 81.34% for the detection task and an accuracy of 66.67% for the classification task. Furthermore, we also compared the performance of GamaComet with an existing free software tool for comet detection, OpenComet. The experiment results showed that GamaComet performed significantly better than OpenComet that could only give an accuracy of 11.5% for the comet detection task. Downstream analysis can be well conducted based on the detection and classification results from GamaComet. The analysis showed that patients owning comet assay images containing comets with class 3 and class 4 had a smoking habit, meaning they had more cells with a high level of DNA damage. Although GamaComet had a good performance, the performance for the classification task could still be improved. Therefore, it will be one of the future works for the research development of GamaComet.

## 1. Introduction

The comet assay or single-cell gel electrophoresis (SCGE) is a simple and precise method to analyze DNA damage [1]. Lymphocytes have been utilized widely in human investigations of DNA damage, but these are difficult to obtain, and the procedure is invasive and may cause discomfort to patients. As a result, researchers began to investigate buccal mucosa cells, which may be obtained through less invasive techniques [2]. Compared to comet assay images obtained from lymphocytes cells, the images produced from buccal mucosa cells contain a lot more noise. Therefore, analyzing comet assay images from buccal mucosa cells is harder than analyzing comet assay images from lymphocytes cells. The buccal cell model can provide essential information on the risk assessment of environmental, occupational, and lifestyle exposures in human biomonitoring research. Buccal mucosa cells are a suitable biomatrix for measuring the degree of personal genotoxicity since they are the first to come into direct contact with substances following exposure to xenobiotics and endogenous damage inductors, such as exposure to dental radiography exams [3,4].

Comet assay analysis can be decomposed into two main tasks. The first task is comet detection from a comet assay image which can consist of some valid comets. In this task, the main objective is judging whether an object in the image is a comet or not. The task is essential since DNA damage is represented as comets in this method. The second task is comet classification which determines the level of DNA damage from the given cell.

Currently, several software tools have been developed for automatic comet detection. Tools such as Comet Assay III [5], Komet version 5.5 [6], Comet Assay Software Project (CASP) [7,8], and a free software, OpenComet [9] can be employed to perform the comet detection automatically. However, the tools cannot be used for the classification task. Hence, the classification is performed manually by experts. Other software tools, CometQ [10] and DeepComet [11] can be used for both comet detection and classification automatically. Nonetheless, the tools are only indented specifically for performing comet assay analysis from lymphocytes cells. Accordingly, the software tool’s performance will drop significantly if the buccal mucosa comet assay images are used which contain a lot of noise. Therefore, a specific software tool is required for automatic comet detection and classification from buccal mucosa comet assay images.

In our previous work, we focused on developing a better tool intended for the comet classification task from comet assay images obtained from buccal mucosa cell using computational methods. First, we used the combination of convolutional neural network (CNN) and transfer learning for the classification task [12]. However, because of the tiny datasets, the performance of CNN was not optimal. Subsequently, we improved the performance of our classifier using a hybrid method of CNN and Extreme Learning Machine (ELM) [13]. The hybrid CNN and ELM increased the performance of the classification since the ELM could minimize the risk of the vanishing gradient problem during the training process. However, in that research, we still had an unsolved problem in which the buccal mucosa comet detection was still performed manually by experts. Therefore, a software tool for fully automated buccal mucosa comet detection and classification is not dispensable.

To solve the problem in our previous work [12,13], in this paper, we propose a deep learning-based tool which can perform both the detection and the classification of comets in each comet assay images obtained from buccal mucosa cell automatically. We refer to Rosati et al. [14] who proposed the usage of a deep learning model, Faster R-CNN [15], to detect and classify comets since the method is superior to other deep learning architectures for both tasks. Even so, we focus on optimizing the deep learning model for comet assay analysis from buccal mucosa cell that is much harder to perform compared to the comet assay analysis from the cell cultures conducted by Rosati et al. [14].

We have implemented our work in a web-based tool, named GamaComet, that can be accessed freely for academic purposes at https://bioinformatics.mipa.ugm.ac.id/gamacomet/. To train and validate the deep learning model in this work, we used data taken from the buccal swab of 24 patients who underwent panoramic radiography for diagnosis and treatment in Prof. Soedomo Dental Hospital, Universitas Gadjah Mada, Indonesia. To test the model, we used data taken from the buccal swabs of seven patients from the same dental hospital. The main contributions of this paper are summarized as follows:We propose a fully automated comet assay analysis tool that can detect and classify comets from buccal mucosa comet assay images. Major improvements are made upon our previous work [12,13] that could only perform the comet assay analysis manually;Our proposed software tool, GamaComet, has been released and can be accessed freely for academic purposes only at https://bioinformatics.mipa.ugm.ac.id/gamacomet/;We used data taken from 24 Indonesian patients to train and validate our proposed deep learning model for GamaComet. Our research tries to tackle the challenge of creating a deep learning model using a small dataset;We also conducted experiments for the testing dataset which had slightly different characteristics from the training and validation datasets. It supposedly can demonstrate the general ability of GamaComet. The testing dataset was taken from seven Indonesian patients;GamaComet can produce better results compared to an existing free tool for comet assay analysis;Downstream analysis can be well conducted based on the detection and classification results from GamaComet.

The remainder of this paper is structured as follows: Section 2 describes the details of the dataset used; Section 3 describes the methodology and the experiment scenario for training the deep learning model; Section 5 provides the discussion of downstream analysis and GamaComet’s performance for the testing data; and Section 6 provides conclusions and future work.

## 2. Dataset

Buccal mucosa samples were collected after radiation exposure from a total of 24 patients in 2018. We had already obtained the ethic committee approval with number KE/FK/0649/EC/2018 related to the sample collection process. The comet assay was performed using an Oxiselect Comet Assay Kit STA-351 (Cell Biolabs, San Diego, CA, USA) with a modified protocol [4]. The comet slide was observed under fluorescence microscopy (Leica, Germany) connected to the optilab and computer. The general outline of the comet assay acquisition process is shown in Figure 1.

This acquisition resulted in the comet assay’s microscope slide image dataset (comet assay images) with a total of 275 images, where each image consisted of some valid comets. All comet assay images acquired were normalized to a size of 1500 × 1125 pixels for reducing the computation and memory load of the deep learning model during the training process [16]. An example of a comet assay image in our dataset is shown in Figure 2. As can be seen in Figure 2, there were five classes (class 0, class 1, class 2, class 3, and class 4) of comet which represent the level of DNA damage of a given cell. Class 0 represents the lowest DNA damage while class 4 represents the highest DNA damage. To develop the deep learning model in this study, the comet assay image dataset was divided into training and validation sets with a ratio of 70% and 30%, resulting in 193 images for the training set and 82 images for the validation set, respectively.

The class distribution of the comet assay image in our dataset is listed in Table 1. We acquired 519 comets from 275 comet assay images. The training set consisted of 193 comet assay images with 365 comets while the validation set consisted of 82 comet assay images with 154 comets.

## 3. Methods

Before proceeding to train a deep learning model, several challenges related to the dataset needed to be tackled in our study. The first one was regarding the number of images that we used for the deep learning model training. Usually, to produce a good and robust deep learning model, a large number of datasets (hundred thousand to millions) are required in the training process [15]. However, in this work, we only had 275 comet assay images. Then came the second challenge in our dataset. The number of comet samples for each class was also very unbalanced. For example, there was a 1:10 difference between class 1 and class 4. Accordingly, we tried to use two kinds of approach to solve these challenges. The first approach was the use of transfer learning in the deep learning model to give a head start in the deep learning model training process [17]. The second approach was the use of data augmentation to increase the number of images in the dataset [18]. Both approaches will be discussed in more detail in the following sections. In this paper, we used Faster R-CNN [15] as it was the current state-of-the-art deep learning algorithm for the object detection and classification task at the time this research was conducted.

### 3.1. Faster R-CNN

Faster R-CNN (Faster Region-Based Convolutional Neural Network) is an improved version of R-CNN [19] and Fast R-CNN [20]. Figure 3 shows the Faster R-CNN architecture used in our study. In this research, we used the Faster R-CNN provided in TensorFlow Object Detection API [21]. The Faster R-CNN architecture consists of Feature Extractor/Backbone Network, Region Proposal Network (RPN), Filters, RoI Pooling, and a Fully Connected Layer. While R-CNN and Fast R-CNN use selective search to generate regional proposals from the input image, Faster R-CNN uses a module called Region Proposal Network (RPN) that works much faster than selective search. RPN works by sliding a small network over a feature map which outputs one or more box coordinate (regression) and the probability of it being an object or not (classification) [15]. This modification in the region proposal method resulted in a much faster network, with an equal or better performance compared to R-CNN and Fast R-CNN. The feature extractor/backbone network will be explained more detail in Section 3.2. Then, by performing max-pooling on the inputs, ROI pooling generates the fixed-size feature maps from non-uniform inputs. ROI pooling layer receive two inputs: (1) A feature map obtained from Feature Extractor/Backbone Network and (2) Region of Interest (RoI) from RPN.

### 3.2. Feature Extractor/Backbone Network

As can be seen in Figure 3, a feature map from feature extractor was used as the input in both the RPN and Region of Interest (RoI) pooling layer. In general, the feature map is usually extracted from the last convolution layer of a Fully Convolutional Network (FCN) that uses the training image as the input. However, the FCN-part of the Faster R-CNN can be altered using an existing CNN-based architecture that has been proven to perform well for extracting features of a given image. Therefore, in this research, we used two variants of ResNet with a different number of layers, ResNet50 [22] and ResNet101 [22], as the feature extractor. The difference between ResNet50 and ResNet101 is that ResNet50 uses a total of 50 convolution layers and ResNet101 uses a total of 101 convolution layers. Performance wise, the combination of Faster R-CNN and ResNet101 scored 32 when tested on the COCO dataset [23] using mean Average Precision (mAP) metric. On the other hand, Faster R-CNN with ResNet50 only scored 30 when tested on the same dataset using the same metric [15].

### 3.3. Transfer Learning

Transfer learning is a technique that focuses on storing knowledge that has been learned from solving a problem in the past and then applies it to another related problem. In the case of object detection using machine learning approach, transfer learning can be performed by training a machine learning model on a specific (source) dataset. Then, using the learned features from the convolution kernels, the model is used to detect images in another (destination) dataset. Several approaches can be used for implementing transfer learning. Moreover, the approaches can be divided based on what model that is used and how it will be used. The model can be made from scratch and then trained on some source datasets until it produces reliable results. Using another approach, we could use the available models from other research studies that had already been pre-trained at some source datasets. Afterwards, we needed to decide whether the model could be directly used to predict the destination dataset or continue performing the fine-tuning for the model by training it more on our destination dataset. The fine-tuning approach is often used when the destination dataset is either smaller or does not resemble the source dataset [17].

### 3.4. Data Augmentation

Data Augmentation is the process of creating new data using existing data. Augmentation that can be performed for image data includes (1) basic image manipulations such as geometric transformations, kernel filters, color space transformations, random erasing, and mixing images, (2) deep learning approaches such as adversarial training, neural style transfer, and GAN data augmentation, and (3) meta-learning such as neural augmentation, auto augment, and smart augmentation [18].

In this research, we used the basic image transformations method, specifically geometric transformations. The method was used to augment the comet assay images, as deep learning methods require a lot of data in the first place to produce a good result. Details on the geometric transformations of our study will be discussed further in Section 3.5.2.

### 3.5. Experiment Scenario

The small size of the comet assay dataset was the main problem that needed to be solved in our study. Thus, in this work, we tried several approaches that have been known to produce or increase the model’s performance when it is used on a small dataset. We designed two experiment scenarios which are described in detail in the following subsections.

#### 3.5.1. Non-Pre-Trained Model vs. Pre-Trained Model

In this scenario, we tested and compared the performance between the pre-trained model and the non-pre-trained model. The transfer learning approach we used in this scenario is listed as follows: for the pre-trained model, we used the Faster R-CNN models that were already trained on the COCO dataset, a large dataset containing around 330,000 images with 1.5 million instances of common objects provided in the TensorFlow object detection API. Afterwards, we fine-tuned the model by continuing the training using our comet assay dataset. By default, the Faster R-CNN model came pre-trained from the TensorFlow Object Detection API. However, with some modifications to the configuration, we could train the Faster R-CNN from scratch using the same architecture as the non-pre-trained model.

#### 3.5.2. Non-Augmented Dataset vs. Augmented Dataset

The augmentation method used in the second scenario was basic image manipulations, specifically geometric transformations. The transformations consist of multiple methods, such as image flipping, image cropping, image rotation, and image translation [18]. In our research, we only used the image rotation method with ±5 degree of rotation. The reason we only used such a small degree of rotation was that the direction of the comets matters in deciding if a comet is valid or not.

For the same reason, we did not use the image flipping method. Image translation and image cropping were also not used in this research as these methods do not really increase the variety of comets in the image unlike the rotation method. We also did not use other basic image manipulation methods such as color space transformation [18] due to the comet assay image itself only containing one color.

To obtain more comprehensive results, we split the second experiment scenario into two sub scenarios with the difference in which data were being augmented. Both sub scenarios were then compared to the model trained on the non-augmented dataset shown in Table 1. Since the objective of the augmentation process was to develop more robust models, the augmentation process was conducted only for training sets.

In the first sub scenario, we considered the class imbalance problem in our dataset. Therefore, we tried to augment the images to balance the dataset. For instance, In Table 1, there was a class imbalance between class ‘3’ and ‘4’ compared to other classes. Thus, we decided to augment those classes to balance the training set. We augmented 42 comet assay images containing class ‘3’ and ‘4’ using the image rotation method with ±5 degree of rotation. After the augmentation process, we had 277 comet assay images in the training set, consisting of 193 original images and 84 augmented images. The comet class distribution after performing the augmentation process is listed in Table 2.

Even so, in the second sub scenario, we tried to augment all classes to increase the overall amount of the training set. Here, the data augmentation of all classes was performed equally without considering the class imbalance. As with the previous sub scenario, we augmented all comet assay images using the image rotation method with ±5 degree of rotation. After the augmentation process, we acquired 579 comet assay images in the training set, consisting of 193 original images and 386 augmented images. Table 3 represents the comet class distribution after all classes were augmented in the second sub scenario.

We mostly used the default configuration that came from the TensorFlow Object Detection API for the Faster R-CNN model, with only slight modifications to some of the parameters related to the dataset, such as the number of classes and the minimum-maximum dimension of the images. For the stopping condition on all models, we decided to stop the training at exactly 100,000 steps while saving checkpoints every 5000 steps.

### 3.6. Evaluation Metric

In this work, we separated the measurement for detection only and detection with classification, as we needed to compare our results with previous tools designed for comet assay analysis. Only comets with a confidence score higher than 0.5 were measured and included in the confusion matrix. Classification accuracy and *F*1-score were then calculated by Formulas (1) and (2).
(1)$$Acuracy=\frac{TP+TN}{TP+TN+FP+FN}$$
(2)$$F1-Score=\frac{2\times TP}{\left(2\times TP\right)+FP+FN}$$
where *TP*, *TN*, *FP*, and *FN* stand for True Positive, True Negative, False Positive, and False Negative, respectively. In the case of detection with classification, a confusion matrix was made for each class in which *TP* represents comet(s) of the current class that is correctly labelled, *TN* represents comet(s) of other classes, which is correctly labelled, *FP* represents comet(s) of other class which is labelled the current class, and *FN* represents comet(s) of current class which is labelled another class.

In the case of detection only, we decided to label previously invalid comets that could not be used as ‘classless’ comets specifically to calculate the detection accuracy of the model. Here, *TP* represents valid comet(s) with classes which are correctly detected, *TN* represents invalid comet(s) which are not detected, *FP* represents invalid comet(s) that are detected, and *FN* represents valid comet(s) with classes which are not detected.

## 4. Results

### 4.1. Non-Pre-Trained vs. Pre-Trained

As shown in Figure 4, the training process conducted on the non-pre-trained Faster R-CNN models showed difficulties in reaching convergence due to the usage of feature extractors with many convolution layers, ResNet50 and ResNet101. Meanwhile, even if the non-pre-trained Faster R-CNN model with ResNet50 (represented as the blue dotted line) as the feature extractor did converge at some points, the pre-trained Faster R-CNN models with COCO dataset (represented as the orange solid line and the blue dashed line) converged much faster compared to the non-pre-trained counterparts. Moreover, we can clearly see that the pre-trained Faster R-CNN models converged at an early step of 10,000. On the other hand, the non-pre-trained counterparts needed more steps to converge. As can be seen in the figure, the non-pre-trained Faster R-CNN model (represented as the green dotted line) did not even converge at all after 100,000 steps.

An example of the predicted output from each model in the first scenario is shown in Figure 5. In the first part of the figure, we show the comet assay input image and its ground truth. Based on the ground truth, there should be five valid comets detected in the given comet assay image: one comet for class 0, one comet for class 1, and three comets for class 2. Moreover, the second part of the figure shows the prediction (detection and classification) results obtained from four different models.

The pre-trained Faster R-CNN model with ResNet50, as shown in Figure 5b, detected and classified more valid comets than other models. Furthermore, the model could appropriately detect and classify four out of five valid comets (one comet for class 0, one comet for class 1, and two comets for class 2). Meanwhile, the non-pre-trained Faster R-CNN model with ResNet50, as shown in Figure 5a, could only appropriately detect and classify three valid comets (one comet for class 0, one comet for class 1, and one comet for class 2). Even though the pre-trained Faster R-CNN model with Resnet 101, as shown in Figure 5d, could detect five comets, the model only classified three valid comets with two false positives. Surprisingly, the non-pre-trained Faster R-CNN model with ResNet101 could not predict anything, as shown in Figure 5c.

Furthermore, the confusion matrix of the classification task from each model can be seen in Figure 6. Since the non-pre-trained Faster R-CNN model with ResNet101 did not predict anything, the confusion matrix in Figure 6c shows zero values for all rows and columns.

The comparison of the evaluation metrics between the non-pre-trained and pre-trained models for the comet detection and the comet classification task are shown in Table 4. In the table, there is N/A for the non-pre-trained Faster R-CNN model with ResNet101 because it did not predict anything. For the comet detection task, the pre-trained Faster-RCNN models achieved a higher overall score than the non-pre-trained Faster R-CNN models. Likewise, for the comet classification task, the pre-trained Faster-RCNN models also achieved a higher accuracy score than the non-pre-trained Faster R-CNN models. Nevertheless, the non-pre-trained Faster R-CNN-ResNet50 model achieved the highest F1-Score (60.18%). After conducting further analysis, we realized that even though the pre-trained model did achieve convergence in early steps, it produced many more False Positives and False Negatives compared to the non-pre-trained Faster R-CNN with ResNet50 late convergence. As a result, the misclassifications reduced the pre-trained Faster R-CNN models’ *F*1-Score. In this scenario, the COCO pre-trained Faster R-CNN model with ResNet50 produced the best overall results, followed by the COCO pre-trained Faster R-CNN model with ResNet101 and non-pre-trained Faster R-CNN model with ResNet50.

### 4.2. Non-Augmented Dataset vs. Augmented Dataset

Since the Faster R-CNN models which were pre-trained with COCO dataset were superior to the non-pre-trained models in our previous experiment scenario, we only conducted experiments using the pre-trained Faster R-CNN models for this scenario. The pre-trained Faster R-CNN models using the non-augmented dataset (the original dataset) were taken from Section 4.1. The Faster R-CNN models trained in this scenario also reached convergence in early steps.

Figure 7 represents the loss comparation chart of all models. There were six models in this experiment scenario: (1) the pre-trained Faster R-CNN with ResNet50 (Faster R-CNN-RN50), (2) the pre-trained Faster R-CNN with ResNet50 using augmented dataset for class ‘3’ and ‘4’(Faster R-CNN-RN50-3&4 Aug), (3) the pre-trained Faster R-CNN with ResNet50 using augmented dataset for all classes (Faster R-CNN-RN50-All Aug), (4) the pre-trained Faster R-CNN with ResNet101 (Faster R-CNN-RN101), (5) the pre-trained Faster R-CNN with ResNet101 using Augmented dataset for class ‘3’ and ‘4’(Faster R-CNN-RN101-3&4 Aug), (6) the pre-trained Faster R-CNN with ResNet101 using Augmented dataset for all classes (Faster R-CNN-RN101-All Aug). From Figure 7, we can conclude that models fine-tuned on comet assay dataset with augmented class ‘3’ and ‘4’ (represented as the blue and purple dotted lines) seem to have reached an overfit stage in the latter steps. The result shows that augmenting a dataset with a low variety will not always increase the performance of a model [18].

Figure 8 represents an example of each model’s prediction from a given comet assay image using the dataset augmentation strategy. As with the previous experiment scenario, the ground truth image contained five valid comets. Here, the pre-trained Faster R-CNN model with ResNet50 and augmented data for class ‘3’ and ‘4’, as shown in Figure 8a, appropriately detected and classified the five valid comets with two false positives. Meanwhile, the pre-trained Faster R-CNN model with ResNet50 and augmented data for all classes, as shown in Figure 8b, appropriately detected and classified three valid comets with one false positive. On the other hand, the pre-trained Faster R-CNN model with ResNet101 and augmented data for class ‘3’ and ‘4’, as shown in Figure 8c, appropriately detected and classified two valid comets with one false positive. Moreover, the pre-trained Faster R-CNN model with ResNet101 and augmented data for all classes, as shown in Figure 8d, appropriately detected and classified four valid comets with one false positive. In addition, the confusion matrix of the classification task from each model trained with the dataset augmentation strategy can be seen in Figure 9.

Moreover, the evaluation metrics comparison is shown in Table 5. The pre-trained Faster R-CNN with ResNet 50 using augmented datasets for all classes (Faster R-CNN-RestNet50-COCO-All Aug.) produced the best results for both the detection and classification tasks. The performance of the Faster R-CNN with ResNet 50 using augmented dataset for class ‘3’ and ‘4’ was worse than the model without the augmented dataset. On the other hand, the performance of the Faster R-CNN with ResNet 101 using the augmented datasets, both the augmented dataset for class ‘3’ and ‘4’ and the augmented dataset for all classes, produced better results than the model without the augmented dataset. From these results, we conclude that Faster R-CNN models with more convolution layers produce better results if they are trained with an augmented dataset.

### 4.3. Faster R-CNN Models vs. OpenComet

In this section, we show the comparison results of an existing free comet assay analysis tool, called OpenComet [9] to our proposed Faster R-CNN models. OpenComet uses conventional segmentation methods for the comet detection task. Since OpenComet was only developed for the comet detection task, we only compared the performance of the detection task in this scenario.

Table 6 shows that the Faster R-CNN models which were fine-tuned with buccal mucosa comet assay images dataset performed better compared to the OpenComet. The result is reasonable since the OpenComet was calibrated more for comet assay images obtained from cell cultures. When we tested the OpenComet on buccal mucosa comet assay images, it produced many False Positives, reaching five times the amount produced by the proposed Faster R-CNN models. Therefore, we can conclude that, compared to conventional segmentation methods used in the OpenComet, machine learning-based detection could perform better in differentiating between objects in a messy environment if there are sufficient data to learn with.

Figure 10 shows an example of the comparison between the output from our proposed model and the output from the OpenComet [9]. Here, we can see that our proposed model appropriately detected three out of five valid comets with only one false positive. On the other hand, even though the OpenComet appropriately detected four out of five valid comets, the tool also detected 20 false positives. Therefore, we can conclude that our proposed model is superior to the OpenComet since it can reduce the false positive rate in the comet detection task.

### 4.4. Implementation of GamaComet

We implemented the proposed Faster R-CNN model as a web-based tool, GamaComet, that can be accessed freely for academic purposes at https://bioinformatics.mipa.ugm.ac.id/gamacomet/. We used the Faster R-CNN with ResNet50 model (which was previously pre-trained on the COCO dataset and fine-tuned on the augmented comet assay dataset), as it produced the best overall results in our experiments.

The system was implemented using Django as a Python web framework, Nginx as a web server, and Gunicorn as an interface between Nginx and the Python application. The basic process of how the system works is shown in Figure 11. A registered user may upload a comet assay image to the GamaComet via a web browser. Afterwards, comets in the given comet assay image will be detected and classified using our model in the cloud server. Finally, the GamaComet will give the detection and classification results to the user.

## 5. Discussion

### 5.1. Downstream Analysis

Our GamaComet used the Faster R-CNN with ResNet50 model (which previously pre-trained on the COCO dataset and fine-tuned on the augmented comet assay dataset), as it produced the best overall results at our experiments. Based on the experiment results as represented at Figure 9b, there were some comets with class 3 and class 4 detected by GamaComet.

Table 7 represents the data of 24 patients involved for collecting buccal mucosa samples to obtain the training and validation datasets of comet assay images. We conducted analysis about the relation between the detected comets and the clinical data of patients. Surprisingly, patients owning comet assay images that contained comets with class 3 and class 4 had a smoking habit (patient number 19 to patient number 24). It can be interpreted that patients with a smoking habit had more cells with high levels of DNA damage (comet with class 3 and class 4), meaning that the downstream analysis could be well conducted based on the detection and classification results from GamaComet.

### 5.2. Performance of GamaComet for Another Dataset

In order to demonstrate the generalization performance of our GamaComet, we conducted an experiment for another dataset (testing dataset). The testing dataset was obtained from buccal mucosa samples collected after the radiation exposure from a total of seven patients in 2016. We had already obtained the ethic committee approval with number 00679/KKEP/FKG-UGM/EC/2016 related to the sample collection process. The testing dataset was obtained using different optilab and different modified protocols with the training and validation dataset. Therefore, the comet assay images from the testing dataset had slightly different characteristics from the training and validation dataset. The testing dataset contained more noises and greener background, represented in Figure 12. Using the testing dataset with different characteristics could demonstrate the general ability of GamaComet.

The testing dataset contained 43 comet assay’s microscope slide images, where each image consisted of some valid comets. The class distribution of the comet assay images in the testing dataset is listed in Table 8. We acquired 73 comets from 43 comet assay images.

GamaComet was run for the testing data. Table 9 represents the accuracy of GamaComet for the testing data. We also compared the performance of GamaComet with OpenComet. Since Open-Comet is only developed for the comet detection task, we only compared the performance of the detection task. GamaComet outperformed OpenComet for the detection task. GamaComet also had a good enough accuracy for the classification task. Although GamaComet had a good performance for the testing dataset both for the detection and classification task, the performance for the testing dataset was slightly less than for the validation dataset, as represented in Table 6. This might have been because the testing dataset had slightly different characteristics from the training and validation dataset. Overall, GamaComet had a good performance for the validation and testing datasets.

## 6. Conclusions and Future Work

In this work, we presented the implementation of the Faster R-CNN model for detecting and classifying comets from buccal mucosa swabs. In our experiments, transfer learning and pre-training helped in training a deep learning model using a small dataset. Not only did it produce an overall better result, but it also requiresd smaller numbers of steps to reach convergence in training. We also concluded that the data augmentation also improved the overall performance of the model, although not significantly. We implemented the proposed Faster R-CNN model, GamaComet, that can be accessed freely for academic purposes at https://bioinformatics.mipa.ugm.ac.id/gamacomet/. We also conducted experiments for the testing dataset with slightly different characteristics from the training and validation datasets. Overall, GamaComet had a good performance for the validation and testing datasets both for the detection and classification task. The detection performance of GamaComet clearly performed better compared to an existing free comet assay detection tool, OpenComet. Downstream analysis could be well conducted based on the detection and classification results from GamaComet. The analysis showed that patients owning comet assay images containing comets with class 3 and class 4 had a smoking habit, meaning that patients with a smoking habit had more cells with high levels of DNA damage.

Although GamaComet had a good performance, the performance for the classification task could still be improved. The classification accuracy of GamaComet for both the validation and testing dataset was less than 70%. Therefore, this will be the next focus for the research development of GamaComet.

For our future works, we intend to improve the performance of GamaComet, especially for the classification task, by increasing the size of our comet assay dataset, as more data clearly improve the performance of a deep learning model. Besides, we also intend to try data augmentation methods that we have not explored yet, such as image mixing or other geometric transformations, adversarial training, and meta-learning.

## Figures and Tables

**Figure 1 diagnostics-12-02002-f001:**
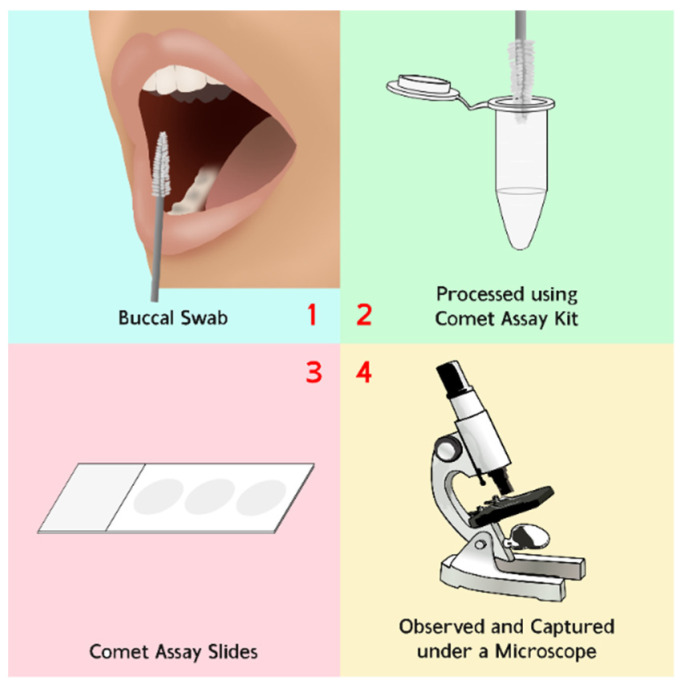
General outline of the comet assay acquisition process.

**Figure 2 diagnostics-12-02002-f002:**
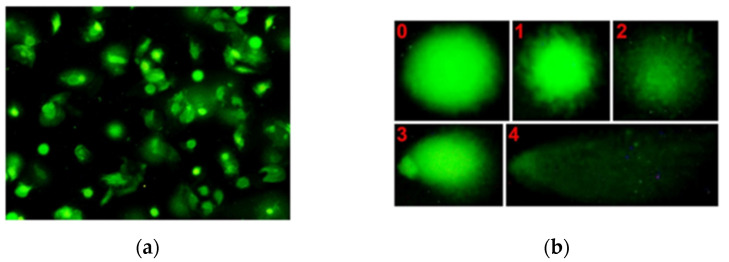
(**a**) An example of a comet assay’s microscope slide image acquired; (**b**) an example of comet classification (five classes).

**Figure 3 diagnostics-12-02002-f003:**
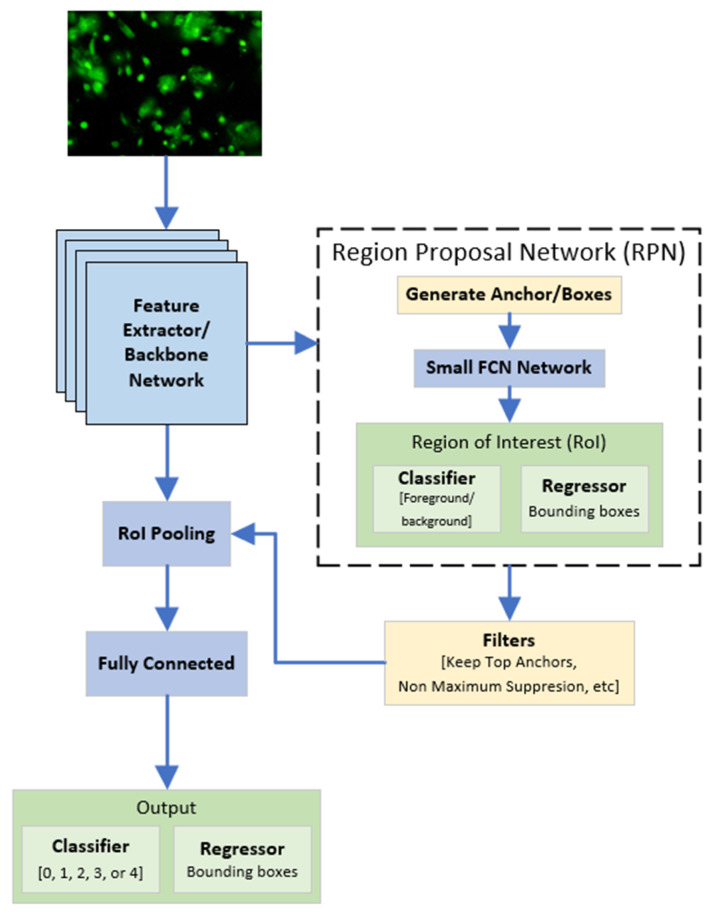
Architecture of the proposed Faster R-CNN for comet detection and classification.

**Figure 4 diagnostics-12-02002-f004:**
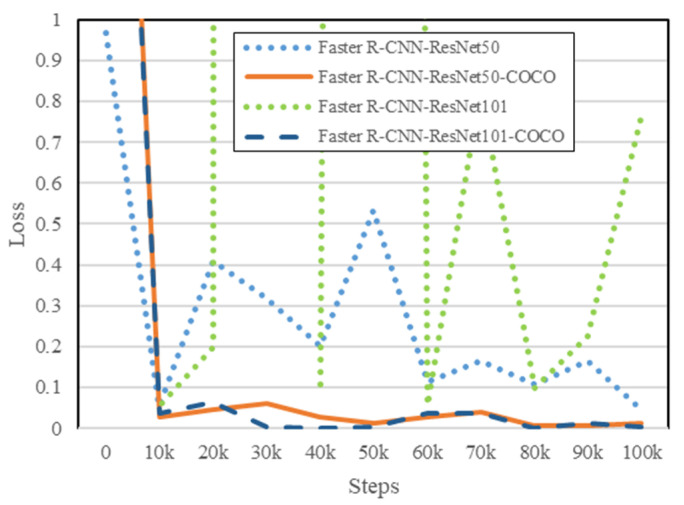
Non-Pre-Trained vs. Pre-Trained: total loss comparation chart.

**Figure 5 diagnostics-12-02002-f005:**
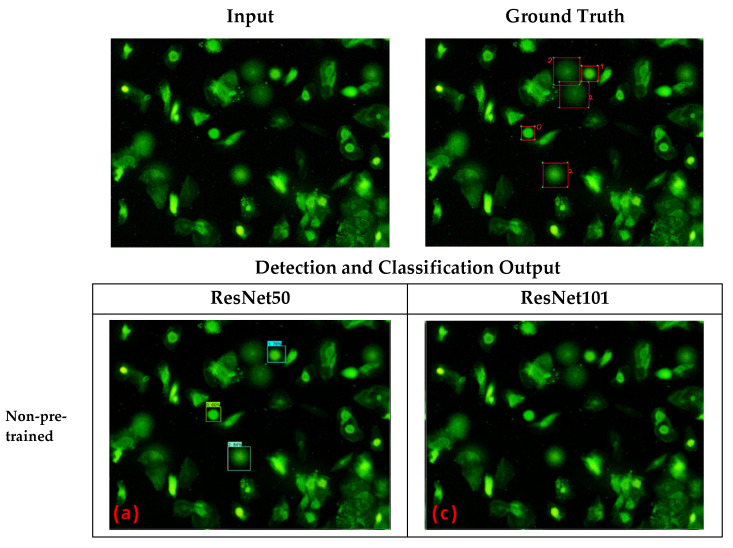
Input Image, Ground Truth, and Non-Pre-Trained vs. Pre-Trained predicted results. (**a**) ResNet50 Feature Extractor: non-pre-trained; (**b**) ResNet50 Feature Extractor: pre-trained; (**c**) ResNet101 Feature Extractor: non-pre-trained; (**d**) ResNet101 Feature Extractor: pre-trained.

**Figure 6 diagnostics-12-02002-f006:**
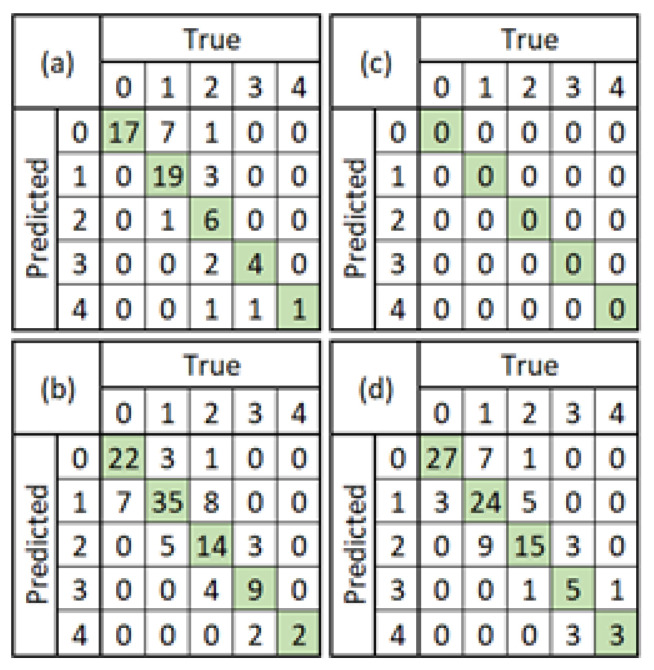
Confusion matrix comparison between the non-pre-trained vs. pre-trained Faster R-CNN models. (**a**) ResNet50 Feature Extractor: non-pre-trained; (**b**) ResNet50 Feature Extractor: pre-trained; (**c**) ResNet101 Feature Extractor: non-pre-trained; (**d**) ResNet101 Feature Extractor: pre-trained.

**Figure 7 diagnostics-12-02002-f007:**
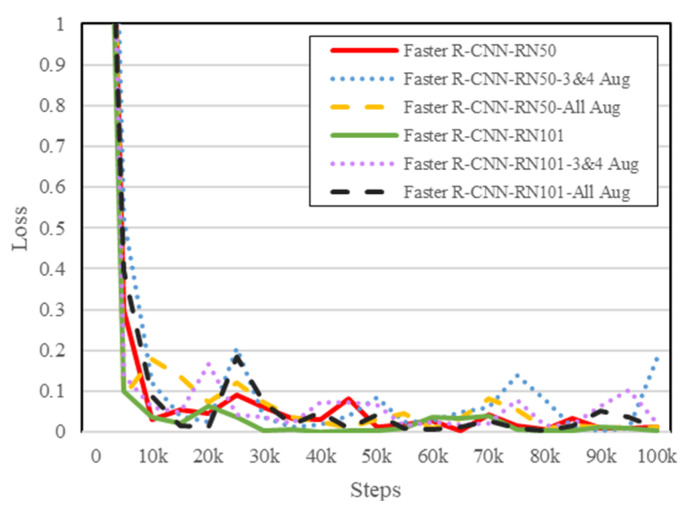
Non-Augmented Dataset vs. Augmented Dataset: Total loss comparation chart.

**Figure 8 diagnostics-12-02002-f008:**
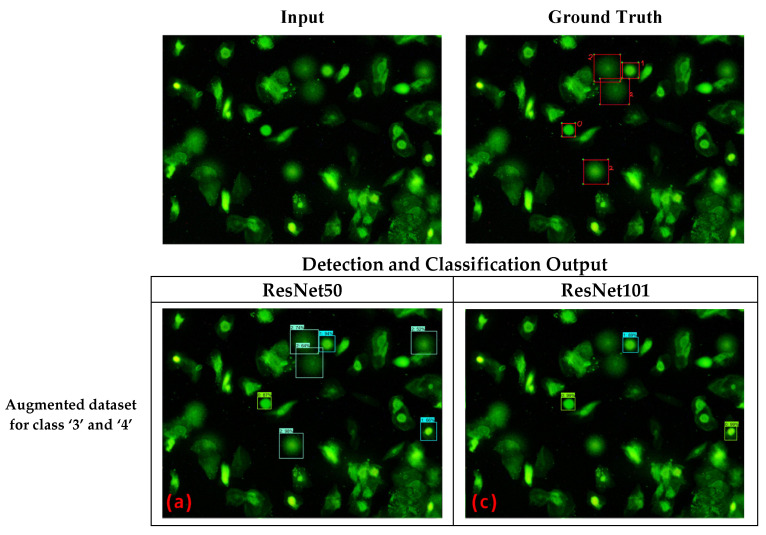
Input Image, Ground Truth, and prediction results of the Pre-Trained Faster R-CNN with the dataset augmentation strategy. (**a**) Faster R-CNN with ResNet50 Feature Extractor: class 3 and class 4 data were augmented; (**b**) Faster R-CNN with ResNet50 Feature Extractor: all classes were augmented, (**c**) Faster R-CNN with ResNet101 Feature Extractor: non augmented dataset; (**d**) Faster R-CNN with ResNet101 Feature Extractor: class 3 and class 4 data were augmented.

**Figure 9 diagnostics-12-02002-f009:**
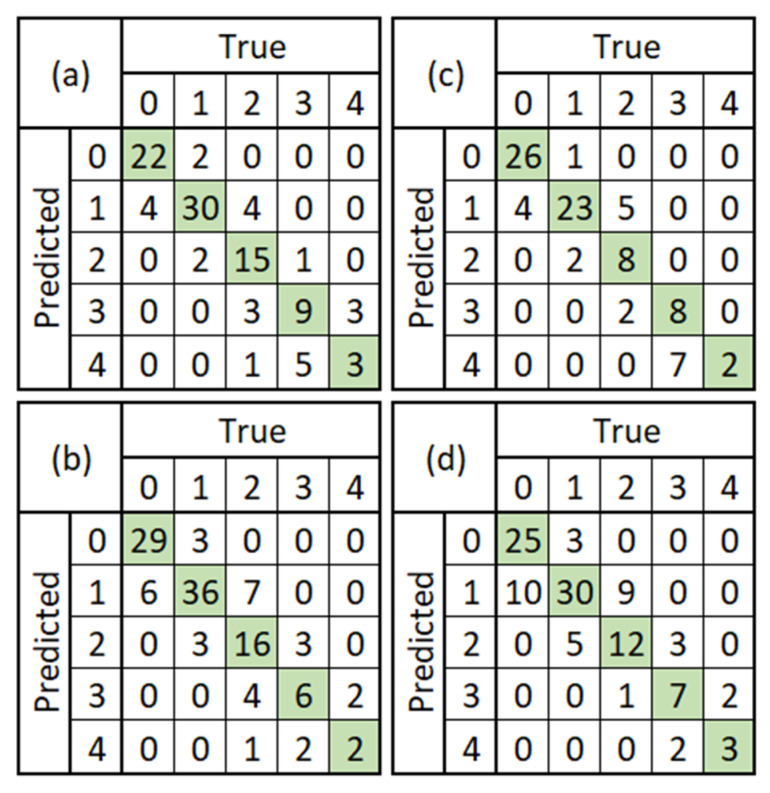
Confusion matrix comparison between 3 and 4 augmented vs. all augmented datasets. (**a**) ResNet50 Feature Extractor: 3 and 4 augmented; (**b**) ResNet50 Feature Extractor: all augmented datasets; (**c**) ResNet101 Feature Extractor: 3 and 4 augmented; (**d**) ResNet101 Feature Extractor: all augmented datasets.

**Figure 10 diagnostics-12-02002-f010:**
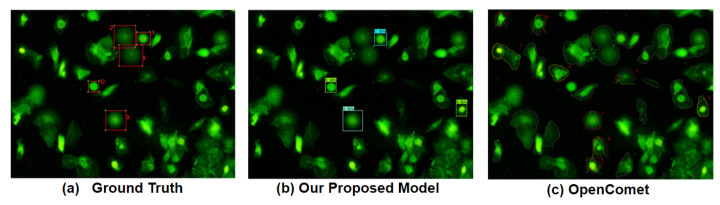
(**a**) A ground truth image; (**b**) the output from our proposed model; (**c**) the output from OpenComet.

**Figure 11 diagnostics-12-02002-f011:**
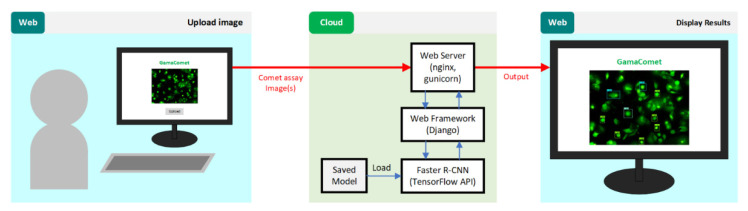
Basic process of the web-based system for comet assay detection and classification.

**Figure 12 diagnostics-12-02002-f012:**
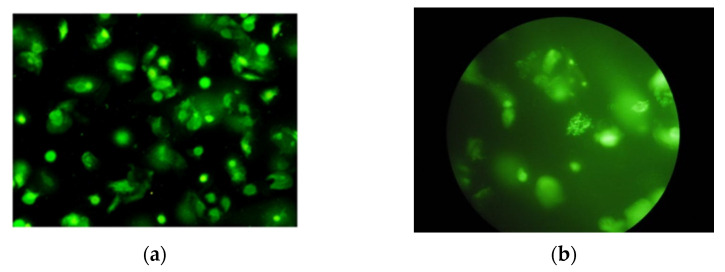
(**a**) An example of a comet assay’s microscope slide image from the training and validation dataset; (**b**) An example of a comet assay’s microscope slide image from the testing dataset.

**Table 1 diagnostics-12-02002-t001:** Details of the comet class distribution in our dataset.

Class	Full Set (275 Images)	Training Set (193 Images)	Validation Set (82 Images)
**0**	127 comets	88 comets	39 comets
**1**	197 comets	143 comets	54 comets
**2**	128 comets	90 comets	38 comets
**3**	48 comets	31 comets	17 comets
**4**	19 comets	13 comets	6 comets
**Total**	519 comets	365 comets	154 comets

**Table 2 diagnostics-12-02002-t002:** Comet class distribution after class ‘3’ and ‘4’ were augmented in the training set.

Class	Training Set		Validation Set (82 Images)
	Before Augmentation (193 Images)	After Augmentation (277 Images)	
**0**	88 comets	88 comets	39 comets
**1**	143 comets	143 comets	54 comets
**2**	90 comets	90 comets	38 comets
**3**	**31 comets**	**217 comets**	17 comets
**4**	**13 comets**	**91 comets**	6 comets
**Total**	365 comets	629 comets	154 comets

**Table 3 diagnostics-12-02002-t003:** Comet assay class distribution after all classes were augmented in the training set.

Class	Training Set		Validation Set (82 Images)
	Before Augmentation (193 Images)	After Augmentation (579 Images)	
**0**	88 comets	264 comets	39 comets
**1**	143 comets	429 comets	54 comets
**2**	90 comets	270 comets	38 comets
**3**	31 comets	93 comets	17 comets
**4**	13 comets	39 comets	6 comets
**Total**	365 comets	1095 comets	154 comets

**Table 4 diagnostics-12-02002-t004:** Comparison of the evaluation metrics between thr non-pre-trained models and pre-trained models.

	Detection		Classification	
	Accuracy	*F*1-Score	Accuracy	*F*1-Score
**Faster R-CNN-ResNet50**	93.66%	62.24%	41.56%	**60.18%**
**Faster R-CNN-ResNet50-COCO**	**95.49%**	**76.54%**	**62.99%**	57.42%
**Faster R-CNN-ResNet101**	N/A	N/A	N/A	N/A
**Faster R-CNN-ResNet101-COCO**	94.51%	70.34%	52.60%	54.47%

**Table 5 diagnostics-12-02002-t005:** Comparison of the evaluation metrics between the non-augmented dataset and the augmented dataset.

	Detection		Classification	
	Accuracy	*F*1-Score	Accuracy	*F*1-Score
**Faster R-CNN-ResNet50-COCO**	95.49%	76.54%	62.99%	57.42%
**Faster R-CNN-ResNet50-COCO-3&4 Aug.**	95.00%	72.31%	55.19%	55.78%
**Faster R-CNN-ResNet50-COCO-All Aug.**	**95.85%**	**78.50%**	**63.64%**	**58.03%**
**Faster R-CNN-ResNet101-COCO**	94.51%	70.34%	52.60%	54.47%
**Faster R-CNN-ResNet101-COCO-3&4 Aug.**	94.45%	71.64%	62.34%	59.42%
**Faster R-CNN-ResNet101-COCO-All Aug.**	95.37%	75.23%	59.74%	57.95%

**Table 6 diagnostics-12-02002-t006:** Comparison of the evaluation metrics between our Faster R-CNN models and OpenComet.

	Accuracy	*F*1-Score
**Faster R-CNN-ResNet50**	93.66%	62.24%
**Faster R-CNN-ResNet50-COCO**	95.49%	76.54%
**Faster R-CNN-ResNet50-COCO-3&4 Aug.**	95.00%	72.31%
**Faster R-CNN-ResNet50-COCO-All Aug.**	**95.85%**	**78.50%**
**Faster R-CNN-ResNet101**	N/A	N/A
**Faster R-CNN-ResNet101-COCO**	94.51%	70.34%
**Faster R-CNN-ResNet101-COCO-3&4 Aug.**	94.45%	71.64%
**Faster R-CNN-ResNet101-COCO-All Aug.**	95.37%	75.23%
**OpenComet** [9]	**76.52%**	**29.87%**

**Table 7 diagnostics-12-02002-t007:** Details of the comet class distribution in the testing dataset.

PatientNo	Sample ID	Collection Date	Sex	Age (Years)	Smoking Habit
1	07092018_1	7 September 2018	Female	24	No
2	07092018_2	7 September 2018	Female	22	No
3	07092018_3	7 September 2018	Female	22	No
4	07092018_4	7 September 2018	Female	27	No
5	07092018_5	7 September 2018	Female	21	No
6	07092018_6	7 September 2018	Male	22	No
7	07092018_7	7 September 2018	Female	22	No
8	13092018_1	13 September 2018	Female	23	No
9	13092018_2	13 September 2018	Female	22	No
10	13092018_3	13 September 2018	Female	21	No
11	13092018_4	13 September 2018	Female	22	No
12	13092018_5	13 September 2018	Female	22	No
13	13092018_6	13 September 2018	Female	22	No
14	13092018_7	13 September 2018	Female	24	No
15	13092018_8	13 September 2018	Female	25	No
16	14092018_1	14 September 2018	Male	18	No
17	14092018_2	14 September 2018	Female	25	No
18	14092018_3	14 September 2018	Female	23	No
19	14092018_4	14 September 2018	Male	23	Yes
20	14092018_5	14 September 2018	Male	24	Yes
21	14092018_6	14 September 2018	Male	25	Yes
22	14092018_7	14 September 2018	Male	24	Yes
23	14092018_8	14 September 2018	Male	20	Yes
24	14092018_9	14 September 2018	Male	23	Yes

**Table 8 diagnostics-12-02002-t008:** Details of the comet class distribution from the testing dataset.

Class	Testing Dataset (43 Images)
**0**	12 comets
**1**	15 comets
**2**	11 comets
**3**	25 comets
**4**	10 comets
**Total**	73 comets

**Table 9 diagnostics-12-02002-t009:** Accuracy comparison between GamaComet and OpenComet for the testing dataset.

	Accuracy	
	Detection	Classification
**GamaComet**	**81.34%**	**66.67%**
**OpenComet** [9]	11.5%	-

## Data Availability

The data used for this research are shared through https://bioinformatics.mipa.ugm.ac.id/gamacomet/ = with permission from authors by send an email to the corresponding author.

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
