# Peer review of "GamaComet: A Deep Learning-Based Tool for the Detection and Classification of DNA Damage from Buccal Mucosa Comet Assay Images"

_diagnostics, 2022, doi:10.3390/diagnostics12082002_

Round 1

Reviewer 1 Report

In this paper, The authors propose a fully automated comet analysis tool that can detect and classify comets from buccal mucosa comet assay images with high accuracy.

Major comments:

1.    Authors should use more datasets to demonstrate the generalization ability of the algorithm.

2.    The authors should add some downstream analysis experiments to demonstrate the significance of the work.

3.    The author should introduce the algorithm in more detail.

Minor comments:

1. The format of this paper should be improved.

2. Authors should improve the innovativeness of the work.

Author Response

Point 1: Authors should use more datasets to demonstrate the generalization ability of the algorithm.
Response 1: Thank you for the suggestion. In the current revised manuscript, we already conducted experiments for another dataset (testing dataset). The testing dataset is obtained from buccal mucosa samples collected after the radiation exposure from a total of 7 patients in 2016. The testing dataset is obtained using different optilab and different modified protocol with the training and validation dataset. Therefore the comet assay images from the testing dataset has slightly different characteristics. Using the testing dataset with different characteristics supposedly can demonstrate the general ability of GamaComet. Overall, GamaComet has good performance for the validation and testing datasets both for the detection and classification task. The performance of GamaComet for the testing dataset is explained comprehensively in the Section 5.2.
Point 2: The authors should add some downstream analysis experiments to demonstrate the significance of the work.
Response 2: Thank you for the suggestion. In the current revised manuscript, we already conducted downstream analysis. The downstream analysis is written comprehensively in the Section 5.1. We conducted analysis about the relation between the detected comets and the clinical data of patients. Surprisingly, patients owning comet assay images that contain comets with class 3 and class 4 have smoking habit. It can be interpreted that patients having smoking habit has more cell with high level of DNA damage. Meaning that the downstream analysis can be well conducted based on the detection and classification results
from GamaComet.
Point 3: The author should introduce the algorithm in more detail.
Response 3: Thank you for the suggestion. In the current revised manuscript, we already explained more detail for the algorithm of GamaComet, which is Faster R-CNN in the Section 3.1.
Point 4: The format of this paper should be improved.
Response 4: Thank you for the suggestion. In the current revised manuscript, we already crosschecked the format of the manuscript following the guidelines from the Diagnostics journal. We restructured the organization of the manuscript. We changed Section 4. Discussion to Section 4. Results and added 1 section, which is Section 5. Discussion. Currently, there are 6 Sections written in the manuscript, which are : Section 1. Introduction, Section 2. Dataset, Section 3. Methods, Section 4. Results, Section 5. Discussion and Section.6 Conclusions and Future Works.
Point 5: Authors should improve the innovativesness of the work.
Response 5: Thank you for the suggestion. We already improved the innovativesness of our work. We conducted downstream analysis and conducted experiments more comprehensively by using another dataset as testing dataset. Overall, we wrote the contributions of our work in the Section 1. Introduction and Section 6. Conclusions and Future Work. We wrote in the Section 1. Introduction, that the main contributions of our work are:
1. We propose a fully automated comet assay analysis tool that can detect and classify comets from buccal mucosa comet assay images.
2. Our proposed software tool, GamaComet, has been released and can be
accessed freely for academic purposes only at
https://bioinformatics.mipa.ugm.ac.id/gamacomet/.
3. We use data taken from 24 Indonesian patients to train and validate our
proposed deep learning model for GamaComet. Our research tries to tackle the
challenge of creating a deep learning model using a small dataset.
4. We also conducted experiments for the testing dataset having slightly different char-acteristics with the training and validation datasets. It supposedly can demonstrate the general ability of GamaComet. The testing dataset is taken from 7 Indonesian patients.
5. GamaComet can produce better results compared to an existing free tool for
com-et assay analysis.
6. Downstream analysis can be well conducted based on the detection and classi-fication results from GamaComet.

Reviewer 2 Report

The main contributions of this paper are summarized as follows: 1. We propose a fully automated comet assay analysis tool that can detect and classify comets from buccal mucosa comet assay images. Major improvements are made upon our previous work  that could only perform the comet assay analysis manually.  The  proposed software tool has been released and can be accessed freely for academic purposes only at https://bioinformatics.mipa.ugm.ac.id/gamacomet/. We use data taken from 30 Indonesian patients. Our research tries to tackle the challenge of creating a deep learning model using a small dataset. The  proposed software tool can produce better results compared to an existing free tool for comet assay analysis.

This paper is structured as follows:

Section 2 describes the details of the dataset used;

Section 3 describes the methodology and the experiments scenario for training the deep learning model;

Section 4 discusses the experiment results and the implementation of the web-based tool, GamaComet;

Section 5 provides conclusions and future work

In spite of the different sections, the paper is difficult to understand in some sections. It uses a computer language, sometimes mathematical, that makes the reader get lost. 

In the abstract it does not give conclusions  

 The organization of the manuscript should have a results section. 

The method should be synthesized. It has at the end of the manuscript conclusions and future but should add limitations of the study. 

Author Response

Point 1: In spite of the different sections, the paper is difficult to understand in some sectons. It uses a computer language, sometimes mathematical, that makes the reader get lost.
Response 1: Thank you for the suggestion. In the current version of manuscript, we avoided to use computer language and mathematical. If it can't be avoided in some parts, we give more explaination so the readers will not get lost.
Point 2: In the abstract it does not give conclusions.
Response 2: Thank you for the suggestion. We already added conclusions in the abstract. We wrote abstract more comprehensively.
Point 3: The organization of the manuscrpt should have a results section.
Response 3: Thank you for the suggestion. In the current revised manuscript, we already crosschecked the format of the manuscript following the guidelines from the Diagnostics journal. We restructured the organization of the manuscript. We changed Section 4. Discussion to Section 4. Results and added 1 section, which is Section 5. Discussion. Currently, there are 6 Sections written in the manuscript, which are : Section 1. Introduction, Section 2. Dataset, Section 3. Methods, Section 4. Results, Section 5. Discussion and Section.6 Conclusions and Future Works.
Point 4: The method should be synthesized. It has at the end of the manuscript conclusions and future but should add limitations of the study.
Response 4: Thank you for the suggestion. We already added limitations of the study in the Section 6. Conclusions and Future Work. We wrote :
… Although GamaComet has good performance, the performance for the classifi-ca-tion task could still be improved. The classification accuracy of GamaComet for both validation and testing dataset are less than 70%. Therefore, it will be one of the next focus for the research development of GamaComet…

Round 2

Reviewer 2 Report

I agree with the changes